# Exploring Maternal Diet-Epigenetic-Gut Microbiome Crosstalk as an Intervention Strategy to Counter Early Obesity Programming

**Maria Felicia Faienza** [1,*], **Flavia Urbano** [2], **Federico Anaclerio** [3,4], **Luigi Antonio Moscogiuri** [2], **Fani Konstantinidou** [3,4], **Liborio Stuppia** [3,4] and **Valentina Gatta** [3,4]

1 Pediatric Unit, Department of Precision and Regenerative Medicine and Ionian Area, University of Bari "A. Moro", 70124 Bari, Italy
2 Giovanni XXIII Pediatric Hospital, 70126 Bari, Italy; flaviaurbano84@gmail.com (F.U.); luigimoscogiuri@hotmail.it (L.A.M.)
3 Department of Psychological Health and Territorial Sciences, School of Medicine and Health Sciences, "G. d'Annunzio" University of Chieti-Pescara, 66100 Chieti, Italy; federico.anaclerio@unich.it (F.A.); fani.konstantinidou@unich.it (F.K.); liborio.stuppia@unich.it (L.S.); v.gatta@unich.it (V.G.)
4 Unit of Molecular Genetics, Center for Advanced Studies and Technology (CAST), "G. d'Annunzio" University of Chieti-Pescara, 66100 Chieti, Italy
* Correspondence: mariafelicia.faienza@uniba.it

**Abstract:** Alterations in a mother's metabolism and endocrine system, due to unbalanced nutrition, may increase the risk of both metabolic and non-metabolic disorders in the offspring's childhood and adulthood. The risk of obesity in the offspring can be determined by the interplay between maternal nutrition and lifestyle, intrauterine environment, epigenetic modifications, and early postnatal factors. Several studies have indicated that the fetal bowel begins to colonize before birth and that, during birth and nursing, the gut microbiota continues to change. The mother's gut microbiota is primarily transferred to the fetus through maternal nutrition and the environment. In this way, it is able to impact the establishment of the early fetal and neonatal microbiome, resulting in epigenetic signatures that can possibly predispose the offspring to the development of obesity in later life. However, antioxidants and exercise in the mother have been shown to improve the offspring's metabolism, with improvements in leptin, triglycerides, adiponectin, and insulin resistance, as well as in the fetal birth weight through epigenetic mechanisms. Therefore, in this extensive literature review, we aimed to investigate the relationship between maternal diet, epigenetics, and gut microbiota in order to expand on current knowledge and identify novel potential preventative strategies for lowering the risk of obesity in children and adults.

**Keywords:** maternal nutrition; epigenetics; microbiome; chronic non-communicable diseases; prevention strategies

## 1. Introduction

Pregnancy is a critical period for fetal growth, particularly vulnerable to inadequate maternal nutrition or adverse intrauterine enviroment, that can program the fetus toward the development of metabolic and nonmetabolic diseases in childhood and adulthood [1–5]. The most important factors that predispose to the development of obesity in childhood are represented by maternal obesity, gestational weight gain (GWG), maternal high-fat (HF) feeding, maternal hyperglycemia, gestational diabetes mellitus (GDM), maternal smoking and stress, low birth weight with rapid postnatal catch-up growth, and high birth weight [6]. Maternal obesity, defined by the body mass index (BMI) either before conception or during pregnancy, and GWG are frequently associated with childhood obesity [7,8].

An elevated maternal BMI during pregnancy and breastfeeding increases fetal adipogenesis in white adipose tissue, while at the same time compromising the development of brown adipose tissue, thus dysregulating fat accumulation and predisposing to obesity later in life [9,10].

According to the most recent literature data, health, or the predisposition to the development of diseases throughout life, is due to a complex and still partially known interplay between exogenous factors and epigenetic modifications that regulate gene expression [11]. Maternal nutrition represents the most important exogenous factor in determining epigenetic changes during pregnancy and breastfeeding. It is well known that some bioactive components of foods can cause DNA methylation, histone acetylation, and modify the availability of enzymatic substrates [12]. The interaction between maternal nutrition and lifestyle, intrauterine environment, epigenetic changes, and early post-natal factors is a determinant for the risk of developing obesity in offspring [13–15]. The accumulation of adipose tissue is also associated with the enhanced expression of Stearoyl-CoA desaturase-1 (Scd1), an enzyme involved in fatty acid metabolism, and reduced Scd1 promoter methylation, supporting the hypothesis that the maternal diet plays a key role in the programming of adipose tissue development through the intrauterine environment [16]. On the other hand, maternal underweight is also a risk factor for intrauterine growth restriction (IUGR) and born small for gestational age (SGA) condition [17]. Children born SGA have a greater risk of developing cardiovascular diseases, obesity, dyslipidemia, type II diabetes, and insulin resistance in adulthood [18–20].

Maternal high-fat diet (HFD) in pregnancy leads to an early accumulation of fat mass, the activation of proinflammatory cytokines, and adipose tissue lipolysis, causing negative effects on adiposity in early life [21].

Maternal hyperglycemia or GDM are considered independent predictors of childhood obesity [22], and the combination of hyperglycemia and obesity has additional effects on the risk of adiposity in childhood [23].

A healthy diet allows a symbiosis between the gut microbiome and the host, with a favorable impact on metabolic health. Studies carried out on fetal meconium suggest that the colonization of the fetal bowel occurs before birth and, subsequently, the gut microbiome undergoes further changes during birth and breastfeeding [24].

The gut microbiome produces metabolites capable of inducing epigenetic modifications and, consequently, modifying gene expression [25].

Evidence shows that newborns born by caesarean section and fed formula milk have a less healthy gut microbiome, unlike vaginal delivery and breastfeeding [26]. A further study demonstrated that newborns born via vaginal delivery have a more various microbiome, rich in Lactobacillus, Bifidobacterial, and Bacteroides, compared to newborns born via cesarean section who show a microbiome in which Clostridium difficile, Clostridium Perfrigens, Staphylococcus, and Streptococcus prevail [27].

In this comprehensive literature review, we aimed to explore the maternal diet-epigenetic-gut microbiome crosstalk to deepen the existing knowledge and to highlight new possible prevention interventions for reducing the risk of obesity in childhood and adulthood.

## 2. Maternal Nutrition and Effects on Fetus and Newborns

Consistent with the origin of health and disease (DOHaD) hypothesis, epigenetic adaptations during intrauterine life occur in response to environmental influences. These adaptations defined as "predictive", if incorrect, can impact the metabolism and increase the risk of chronic diseases in adulthood [28]. The term "developmental programming" refers to the molecular, cellular, neuroendocrine, physiological, and metabolic pathways that can be modified because of an excess or deficiency of nutrients, hormonal alterations, stress, and placental dysfunction.

During pregnancy, the nutritional needs of both the mother and the fetus depend on the gestational period. During the early stages of pregnancy, the maternal diet does not seem to have an impact on the weight of the newborn, while in the final stage of pregnancy, a HFD (>35% of energy intake) can alter the body composition of the fetus and predict the adiposity [29]. Maternal over or under nutrition during pregnancy may lastingly alter gene expression through epigenetic mechanisms, leading to metabolic alterations and obesity programming in prenatal life [30]. It has been demonstrated that the tight adherence to the Mediterranean diet during pregnancy reduces the risk of overweight/obesity in offspring at 4 years [31], although other studies did not find any association between a maternal diet and fetal and childhood adiposity [32,33]. Moreover, specific nutrients, such as Docoexhaenoic acid (DHA), in obese women or in women with GDM administered in late pregnancy have been positively associated with lower adiposity in infants under exclusive breastfeeding [34].

Maternal excess weight gain during pregnancy can be due to the preference for HFD and high-sugar diets (HSD) in the gestational period. This condition may alter metabolism and fetal body composition and, by increasing leptin expression in subcutaneous and visceral fat [30]. In a systematic review and meta-analysis, maternal HFD resulted correlated with high body fat, high leptin, glucose, insulin, and triglycerides levels, as well as hypertension later in life [35]. Maternal HFD has been shown to be associated with alterations in the hypothalamic regulation of weight and energy homeostasis in the fetus, and with changes in eating behavior after birth, through the regulation of the expression of the gene encoding for the leptin receptor, POMC and neuropeptide Y [36,37].

Furthermore, maternal obesity and HFD during pregnancy are also associated with hyperlipidemia and insulin resistance which increase adipose tissue lipolysis causing a substantial release of free fatty acid (FFA) in the fetus, and the activation of inflammatory cytokines [21]. The inflammation and plasma FFAs can modify the normal development of fetal organs such as brain, skeletal muscle, adipose tissue, liver, and pancreas rising the incidence of metabolic alterations [37]. The prevalence of obesity among adolescents born in obese mothers and in mothers with GDM is, respectively 40% and 26% [38]. The risk of childhood obesity has been significantly associated with excess of maternal weight gain during gestation [39]. A study conducted in 609 mother-child dyads followed for 36 months postpartum showed that the increased risk of childhood obesity is associated with GWG [40].

Maternal undernutrition can lead to intrauterine growth restriction (IUGR) and metabolic alterations later in life. The Dutch famine study demonstrated a link between the gestational period of maternal undernutrition, the metabolic environment of the fetus, and the subsequent risk of developing obesity, suggesting the presence of critical windows of susceptibility [41]. Experience of maternal undernutrition during the first and second trimesters leads to an increase in the prevalence of obesity and cardiometabolic risk in offspring, compared to those exposed in the third trimester. Maternal undernutrition may influence the development of obesity through a "predictive adaptive response" secondary to reduced nutrient availability in the uterus, in which the fetus programs itself in response to a persistent caloric deficit. However, the response becomes maladaptive in the context of a mismatch between the expected environment and nutritional exposures across the lifespan. Maternal undernutrition can therefore influence a "thrifty phenotype", with the result that offspring develop obesity later in life following exposure to a high-calorie diet [42].

## 3. Maternal Dietary Factors Influencing Metabolic and Endocrine Changes

Pregnancy represents a window of opportunity to plan the future health of both mothers and newborns. Indeed, the endocrine and metabolic changes occurring during pregnancy have an impact on the long-term risk of chronic diseases, through a transgenerational flow theorized by the DOHaD theory [28]. Several mechanisms can determine a transgenerational programming of metabolic traits through the maternal line, including

epigenetic effects via the germline, intrauterine environment, somatic epigenetics, and mitochondrial programming [43].

The pregnancy is characterized by a complex metabolic and endocrine adaptations related to both the pre-gestational nutritional status, gestational diet, and gestational weight gain. These modifications play a crucial role in appropriate fetal development. Several molecules as glucose, fatty acids, hormones, and adipokines require a correct balance, in which the placenta has a key role [44]. In addition, the placenta acts as endocrine organs, secreting adipokines such as resistin, chemerin, omentin, free fatty acid binding protein-4, and retinol binding protein-4, which modulate metabolism and insulin resistance (IR) in pregnancy [45,46]. Maternal nutrition in turn regulates placental development and function by epigenetic modulation of several amino acid and glucose transporters [47,48]. The deficiency or excess of maternal circulating nutrients and the consequent effects on the distribution of fetal nutrients across the placenta represents the basis of fetal programming. The first phase of gestation, defined as the "maternal anabolic phase", is characterized by high maternal energy resources, mainly as lipid deposits, to support the maternal-fetal needs of late gestation and breastfeeding. The second phase, called the "maternal catabolic phase" or "fetal anabolic phase", serves for the fetal growth. This phase is characterized by a significant maternal IR which is critical in supporting the growing fetus during the third trimester [49,50]. Despite a progressive decrease in fasting glucose, and a 3.0 to 3.5-fold increase in fasting insulin, liver glucose production through gluconeogenesis and glycogenolysis represents an adaptation of maternal metabolism to the growing carbohydrate needs of the fetus and placenta [51]. Maternal nutrition and metabolic status change also lipid transport through the placenta [52,53]. During the anabolic phase, the increase in estrogen and progesterone levels stimulate lipid accumulation [44]. In the third trimester, the transition to the catabolic state occurs and further deposition of fat mass stops, and lipids become the main source of energy, while glucose and amino acids are stored for the fetus [54]. The placenta protects the fetus from an adverse metabolic environment, e.g., through the accumulation of fatty acids in lipid droplets; however, when adverse metabolic conditions develop, such as severe maternal obesity or severe hypertriglyceridemia, the placenta is no longer able to accumulate lipids, resulting in an overflow to the fetus [55]. On the other hand, in fasting status, a fast change of maternal metabolism towards lipid oxidation occurs, with both an increase in FFA and the production of ketones. This influx of ketone bodies from maternal to fetal circulation ensures embryonic brain development in condition of nutrient insufficiency. However, maternal hyperketonemia could have negative effects on the fetus, such as fetal malformations and altered neurophysiological development [50,56]. Maternal diet can expose the offspring to glycotoxins, which are the result of hyperglycemia or fast dry-heat cooking. Glycotoxins include advanced glycation end-products (AGEs), and their precursors, such as methyl-glyoxal (MG) which transforms the arginine and lysine residues of proteins and DNA, forming AGEs [57–60]. The extracellular AGEs may trigger membrane receptors, such as RAGE, which activate oxidative stress and inflammatory signaling [61]. These mechanisms are involved in the senescence of endothelial cells, podocytes and neurons, and diabetic complications (retinopathy, nephropathy, and peripheral neuropathy) [57], and in the pathophysiology of cardiovascular, cerebrovascular, and degenerative diseases.

Maternal diabetes and/or obesity stimulate fetal insulin secretion, and several anabolic processes such as lipogenesis, protein synthesis, and fetal glucose consumption in insulin-sensitive cells as hepatocytes, myocytes, adipocytes, causing expanded fetal growth, increasing glucose requests and IR in late gestation [62]. During brain development, hyperinsulinemia can alter metabolic and energy regulation due to the programming of the hypothalamic metabolic regulation centers [63]. Indeed, maternal hyperglycemia, due to diabetes or obesity, has been linked with an increased incidence of obesity and diabetes among the offspring later in life [64].

Leptin is a hormone which controls body weight and several other processes, such as the immune and the systemic inflammatory response. Consequently, leptin can act as a metabolic switch, linking the nutritional status to high energy-consuming processes. In pregnancy, leptin is produced also by the placenta and its circulating levels gradually increase during gestation for maintaining energy intake for fetal growth, particularly in the second and third trimesters [65]. In obese pregnant women, hyperinsulinemia stimulates leptin production which in turn augments inflammation [66]. This amplified pro-inflammatory response, together with the increase in TNF-alpha levels, and the enhanced production of reactive oxygen species (ROS), result in IR in the feto-placental vascular unit which can determine diseases such as GDM and preeclampsia [67–69]. Thus, maternal obesity, excessive gestational weight gain, and gestational diabetes may be associated with a state of "meta-inflammation" that configure a metabolic condition recently indicated as "Gestational Diabesity" [70,71].

On the other hand, low maternal glucose levels due to maternal undernutrition leads to IUGR [72]. In this condition, fetal brain development is favored through fetal blood re-distribution, while other organs, especially the kidney and liver, become exposed to growth restriction [73,74]. Furthermore, maternal undernutrition and low fetal glucose availability can result in decreased pancreatic β-cell mass with consequent insulin hyposecretion, potentially leading to an augmented susceptibility to diabetes in adult offspring [75,76].

The role of GCs during pregnancy is crucial for fetal survival. Glucocorticoids (GCs), synthetized by the adrenal glands under the control of the hypothalamic-pituitary-adrenal (HPA) axis, preserve homeostasis by regulating energy metabolism, inflammation and other biological processes [77]. The maternal HPA axis undergoes adjustments during pregnancy [78]. At the beginning of pregnancy, GCs are essential for the embryo implantation. During gestation, the maternal adrenal glands gradually become hypertrophic due to placental secretion of corticotrophin releasing hormone (CRH), and consequently, a peak in cortisol concentration occurs in the third trimester, leading to a 3- to 8-fold increase in total serum levels [77,79]. Concomitantly, the fetal adrenal glands temporarily produce cortisol from approximately 7 to 10 weeks of gestation.

The placenta also regulates fetal cortisol levels through the expression of 11β-hydroxysteroid dehydrogenase 2 (11β-HSD2), which converts cortisol to its inactive form, cortisone. The "window" and duration of GC exposure during gestation are precisely regulated and any alterations can lead to pregnancy complications and the development of permanent diseases [80]. In the presence of malnutrition, stress, and infections, the HPA axis may undergo overstimulation, with subsequent 11β-HSD2 dysregulation, and fetal HPA axis hyperactivity. These alterations could lead to preeclampsia, IUGR, hypertension, and life-long outcomes caused by early metabolic stress and potential epigenetic alterations [81].

Despite the importance of the growth hormone (GH) in fetal development, there are few data on the programming effects of maternal and/or fetal GH on the hypothalamic development of offspring. Experimental data have demonstrated a regulation of GH on the formation of POMC and AgRP axons [82]. Further studies are necessary to establish if altered maternal or fetal GH levels have an effect on the development of the hypothalamus with alterations, causing obesity.

## 4. Maternal Dietary Factors Influencing Epigenetic Changes

There is substantial evidence to support the short- and long-term health advantages of a healthy diet and lifestyle for women prior, during, and after pregnancy. Pregnancy increases the body's need for nutrients in order to support the growth and development of the fetus as well as the mother's new tissue and metabolism. Optimal dietary intake during the postpartum period is also crucial to support the extra nutritional needs for breastfeeding after gestation [83]. Early nutritional exposures in infancy can impact normal development and increase the risk of childhood obesity and metabolic programming later in life [84].

Furthermore, breastfeeding can also influence the growth, cognitive development, and microbiota of the newborn. In particular, bioactive molecules, such as human milk oligosaccharides (HMOs), have been linked to the acquisition of various developmental milestones, but the extent to which they have an impact depends on maternal secretor status [85].

Epigenetics consists of a vast array of heritable biological modifications able to modify gene expression or activity without altering the DNA sequence. The main epigenetic changes are represented by DNA methylation, the expression of non-coding RNAs (ncR-NAs), and histone modifications [86]. Nutrition is considered essential for regulating epigenetic processes [87]. The human epigenome is highly malleable during fetal and early life development and is vulnerable to environmental influences. As it has been shown in the pancreas of murine models, a low-protein maternal diet during pregnancy may limit the availability of nutrients, leading to alterations in the expression and methylation status of key pancreatic genes, such as Pdx1 and MafA, and to the consequent developmental dysregulation of β-cell activity with long-term health consequences on the offspring [88]. Hence, maternal dietary habits throughout pregnancy, as well as the early years of life may be a significant impact factor for the offspring's epigenetic programming. Thus, compromised maternal nutritional choices can result in epigenetic changes able to negatively influence the development of the embryo or even the epigenetic profile of the offspring at birth [89].

In particular, maternal nutrition can specifically affect DNA methylation through the one carbon cycle, a metabolic pathway reliant on micronutrients like methionine, folate, and vitamins B6 and B12 with potential effects on fetal programming [90]. Leptin has also been examined in detail because of its central role in the regulation of food intake behaviors, metabolism, and inflammation [91], and because of the strong association between the regulation of its expression levels and epigenetic modifications [92].

Bisulfite pyrosequencing experiments have been performed in mothers and their healthy neonates in order to evaluate the methylation of CpG sites in the promoter region of the placenta leptin (*LEP*) gene and ascertain the genotype of the rs2167270 *LEP* single nucleotide polymorphism (SNP), which is known to influence leptin methylation, in association with the maternal dietary habits during the gestational period [93]. It was demonstrated that the rs2167270 (+19G > A) genotype was an important predictor of placenta leptin DNA methylation, and that lower levels of this methylation were associated with a higher intake of added sugars and white or refined carbohydrates. These results could highlight the importance of carbohydrate consumption on the *LEP* gene methylation in the placenta. Lower methylation may be a sign of a placenta response to high calorie and carbohydrate foods, which would raise this hormone's levels during fetal development and impact infant growth, especially considering that methylation decreases gene transcription. Furthermore, a meta-analysis of epigenome-wide association studies (EWAS) in mother–child pairs on maternal adherence to the Mediterranean diet during pregnancy and cord blood DNA methylation of offspring was also performed [92]. It was shown that maternal adherence to the Mediterranean diet throughout pregnancy was associated with cord blood DNA methylation at a specific CpG, called cg23757341, which corresponds to the transcription start site of the WNT5B gene. A negative association of DNA methylation at cg23757341 with fasting insulin in adult whole blood [94], and a connection between overexpression of WNT5B and increased adipogenesis in adipocytes, compatible with a role in the development of type 2 diabetes, have already been established. Therefore, these findings could be considered significantly supported and able to highlight a potential cause-related mechanism behind maternal Mediterranean diet and offspring DNA methylation patterns.

The Dutch Hunger Winter Families study has also been thoroughly investigated [95–98]. Throughout the winter of 1944–1945 the Dutch famine, caused by a food transport embargo of the German military forces, offered a rare chance to investigate the impact of various stages of gestational undernutrition on adult health [99]. In recent years, Tobi et al. [100] identified singleton births between 1945 and 1946 at three institutions in the western Netherlands whose mothers were exposed to this famine during or immediately prior to their pregnancy, selecting same-sex siblings not exposed to the famine as sibling controls. DNA methylation analysis was performed in the whole blood of in 422 individuals exposed to famine in utero and 463 sibling controls, in order to better elucidate the relationship between adverse intrauterine environment and adult metabolic health. Part of the results evidenced that DNA methylation at CpG cg09349128 of the PIM3 gene, involved in energy metabolism, mediated 13.4% of the association between famine exposure and BMI, while DNA methylation at CpGs of TXNIP, influencing β cell function, and ABCG1, impacting lipid metabolism, together mediated 80% of the association between famine and triglycerides, strongly suggesting that DNA methylation may be considered to be a mediator in the association of prenatal famine exposure with higher adult BMI and serum triglyceride levels.

Nonetheless, nutrition, also in the postnatal period, may be associated with metabolic programming. One of the potential underlying mechanisms equally involves epigenetic modifications, with particular reference to DNA methylation. Because human milk contains the perfect balance of nutrients, hormones, like leptin, and other factors, like immunoglobulins, which are essential for growth and health, breastfeeding is protective against a number of chronic diseases. However, its effect on DNA methylation at an early age remains unclear [101].

The Maternal Nutrition and Offspring's Epigenome (MANOE) study assessed the effect of breastfeeding duration on infant growth and DNA methylation in obesity-related genes by comparing buccal epithelial cells obtained from children who were breastfed for less than six months and children who were breastfed for more than six months [102]. A significant difference was observed between infant growth and buccal retinoid X receptor alpha (*RXRA*) and *LEP* gene methylation at 12 months of breastfeeding. Infant weight was also found to be significantly lower when children were breastfed for 10 to 12 months, evidencing that breastfeeding duration was also linked to infant biometry. Therefore, this study suggested that breastfeeding can impact infant growth and offspring buccal DNA methylation levels in fat metabolism (*RXRA*) and appetite regulation-related (*LEP*) genes, and that extended breastfeeding may have an impact on the development of childhood obesity, which could be accounted for by an upregulation of RXRA and a downregulation of LEP in one-year old children.

The duration of breastfeeding has also been evaluated in reference to *LEP* DNA methylation profiles and BMI in peripheral blood of 10-year-old children [103]. Subsequently, this research also concentrated on the association between breastfeeding exposure and BMI trajectories covering the first 18 years by using multi-nominal logistic regression. It was evidenced that at several *LEP* CpG sites, the length of exclusive and total breastfeeding was linked to DNA methylation at 10 years, but not at 18 years. The duration of breastfeeding was also associated with the early transient overweight trajectory. Moreover, the DNA methylation of *LEP* was correlated with this trajectory at one CpG site and early persistent obesity at another. As a result, a significant association of breastfeeding duration with altered methylation at a total of seven CpG sites on the *LEP* gene was determined and a further correlation of breastfeeding duration with childhood obesity was confirmed. Although it has been widely proposed that epigenetic mechanisms may explain the associations between breastfeeding and long-term health outcomes in children, there is currently still little evidence in the literature to support these theories and, therefore, a need to investigate this specific issue in a more detailed manner.

### 5. Maternal Diet–Epigenetics–Microbiome

The maternal dietary choices are central in providing the essential components for the developing fetus during pregnancy. In a fetal and neonatal developmental context, an increasing number of studies are emphasizing not just the significance of the maternal diet, but also the impact of epigenetics and the establishment and composition of the gut microbiome. Dietary regime is a crucial factor for the composition of gut microbiota, facilitating communication with the intestines, and the transmission of signals to peripheral organs. Changes in dietary habits can cause shifts in the composition and functionality of microbial communities, leading to alterations in various processes such as fermentation, energy metabolism, permeability, and sensation [104,105]. Additionally, recent evidence indicates that diets significantly influence epigenetic mechanism associated with obesity development [106].

Maternal dietary patterns or nutritional compositions play a crucial role in shaping the epigenetic profiles of the fetus. The impact of the maternal diet on the health of offspring is an area of research closely tied to epigenetics. For example, the major metabolic activity involves synthesis of metabolites [107] such as vitamin B12, folate, choline, and betaine. These compounds potentially contribute to the production of 6-methyltetrahydrofolate, a methyl group donor crucial for generating S-adenosylmethionine (SAM) that participates in DNA methylation processes [108–112]. The regulation of these methyl donor nutrients are found to be regulated by specific gut microbial communities, such as *Lactobacillus* and *Bifidobacteria,* which are known for folate production [113]. To comprehend the role of *Lactobacillus* and *Bifidobacterium*, a study by Murri et al. revealed a decrease in their levels among healthy Caucasian children with type 1 diabetes [114]. These findings underscore a significant connection between gut microbiota and DNA methylation mechanisms that could potentially govern diabetes. Gut microbiota represents a host environment and is also considered as "the second genome" [115], with trillions of microorganisms, containing about 1000 bacterial species in particular [116]. Advancements in gene sequencing have revealed that the genome of human gut microbial communities, comprising approximately 3 million genes, is over 100 times larger than the human genome. This is particularly noteworthy considering that the ratio of human to bacterial cells in the gut is estimated to be approximately 1:1 [117]. Primarily, the gut microbiota is populated by the Phila *Bacteroidetes*, *Firmicutes*, Actinobacteria, and *Proteobacteria*. *Firmicutes* and *Bacteroidetes* constitute 90% of all bacterial species. The most abundant phylogenetics class are *Clostridia*, *Bacteroidia*, and *Negativicutes*, meanwhile, the major genera are *Bacteroides*, *Clostridium*, *Faecalibacterium*, *Streptococcus*, *Pseudomonas*, *Prevotella*, *Fusobacteria*, *Veillonella*, *Neisseria*, *Porphyromonas*, *Eubacterium*, *Ruminococcus, and Bifidobacterium* [118,119]. Several elements are pivotal in shaping the gut microbiome of an infant. Primarily, the gut microbiome is acquired from the mother during gestation through maternal nourishment and the embryonic milieu. Furthermore, dietary habits play a fundamental role in the development and preservation of the gut flora, including breastfeeding, formula consumption, and dietary composition. Additional factors influencing its formation encompass genetic predisposition and environmental variables, like pharmaceutical or antibiotic usage, illnesses, lifestyle choices, relocation to alternative settings, and more. Various research investigations have indicated that infants delivered via vaginal birth exhibit an initial and plentiful presence of *Lactobacillus*, *Bacteroides*, and *Prevotella*. Conversely, infants delivered via cesarean section experience a postponement in the appearance or reduced quantities of *Bacteroides*, *Bifidobacteria*, and *Lactobacillus*, with a predominant colonization of *Clostridium difficile* and *Escherichia coli* [120–123]. These species are recognized as probiotics because of their beneficial effects on health and their preventative role in various metabolic conditions like obesity and diabetes [124] (Figure 1).

Given that it seems probable that the crucial bacteria obtained by infants born vaginally originate from the maternal fecal pool, one method to establish a healthy bacterial population in the infant gut following cesarean delivery has been through fecal microbiota transplantation (FMT) using the mother as a donor. In the study by Korpela et al. they evaluate the FMT approach in 17 mothers. This approach has been demonstrated to furnish infants delivered via cesarean section with microbiomes similar to those observed in infants born vaginally, highlighting the importance that the most pivotal period for the development of the human microbiome occurs during the initial 2–3 years of life. Addressing dysbiosis during pregnancy and early childhood can profoundly impact the future health of an infant [125].

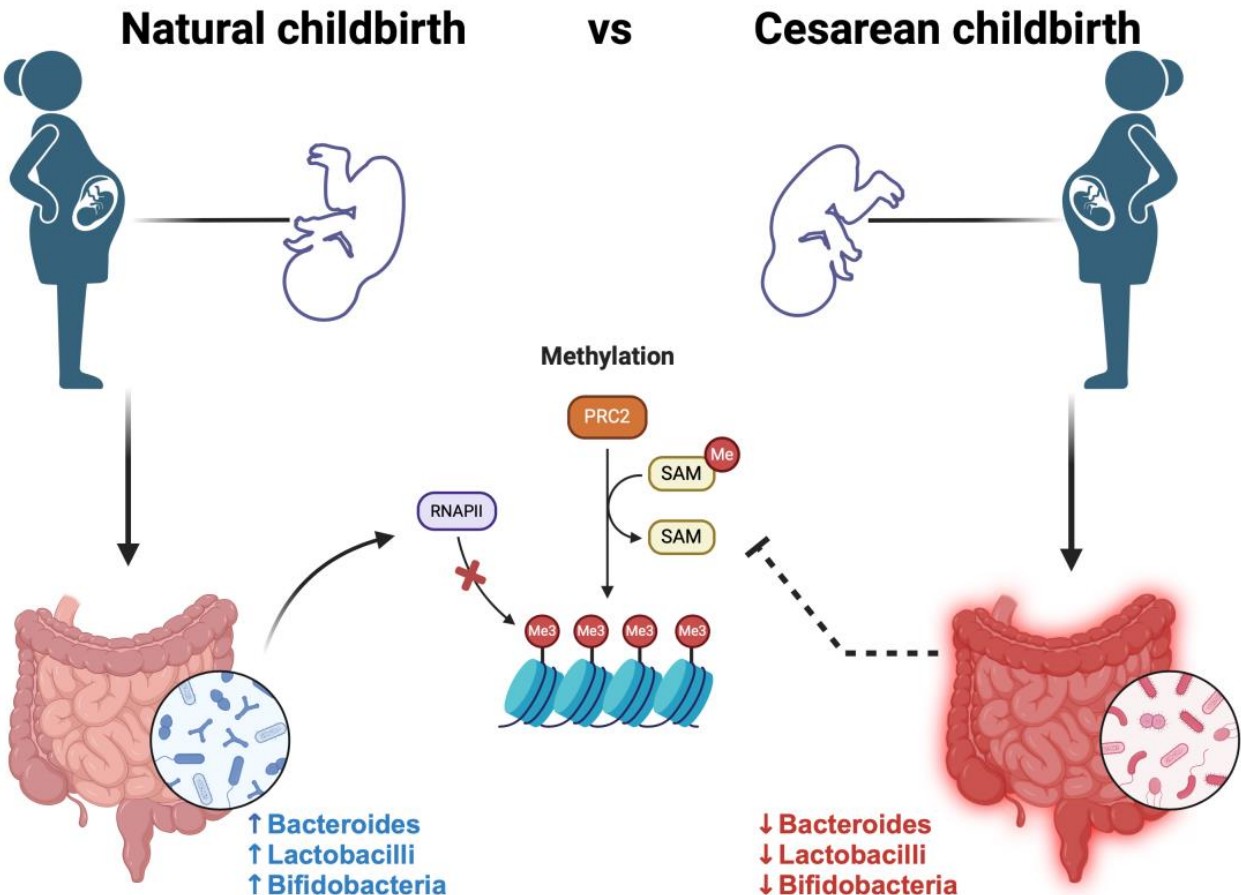

**Figure 1.** Different studies suggest that infants delivered via natural childbirth exhibit an initial and better gut microbiota for the presence of *Lactobacillus*, *Bacteroides*, and *Prevotella*. These bacteria are known to participate in DNA methylation processes, due to their role as methyl group donors for the production of nutrients such as vitamin B12, folate, betaine, and choline, which, for example, are potentially involved in the synthesis of 6-methyltetrahydrofolate, for the generation of S-adenosylmethionine (SAM). Conversely, infants delivered via cesarean section experience a postponement in the appearance or reduced quantities of *Bacteroides*, *Bifidobacteria*, and *Lactobacillus*, with a predominant colonization of *Clostridium difficile* and *Escherichia coli*.

While the therapeutic efficacy of fecal microbiota transplantation (FMT) has been well-established in humans through various studies, its application in animal models, particularly in dogs, has been less explored. Only a few studies have investigated the use of FMT in dogs. One of these studies highlighted the benefits of FMT, demonstrating increased fecal bacterial diversity and a metabolome more closely resembling that of healthy controls. Additionally, FMT shows potential as a therapeutic approach for dogs suffering from

chronic intestinal diseases and related dysbiosis, such as various chronic enteropathies and exocrine pancreatic insufficiency [126].

These evidences could suggest that interventions to modify the maternal microbiome could actually prevent adverse epigenetic changes and modify the phenotype of the fetus. In these studies, fecal transplants have shown that improvements in the maternal microbiome have beneficial effects on maternal metabolism and on fetal birth weight and microbial diversity.

Numerous investigations have highlighted that shifts towards an unfavorable microbial balance are primarily linked to the consumption of a "Western" diet, characterized by high-fat and carbohydrate content. Kumar et al. observed that infants born to mothers with a higher proportion of *Firmicutes* in their gut exhibited modified DNA methylation patterns compared to infants born to mothers with a higher proportion of *Bacteroidetes* [82]. These microbial communities are known to directly impact epigenetic changes through DNA methylation. In addition to dietary influences, other factors, such as antibiotic usage and infections, can also affect the gut microbiota [127], leading to imbalances and decreased microbial diversity, a condition referred to as gut dysbiosis. Generally, the first aspect of gut dysbiosis is an unbalanced percentage relationship between the most important Phila, such as *Firmicutes* and *Bacteroidetes*. In particular, these two Phila in normal conditions stay in a 1:1 ratio, but when there is a particular dysbiosis, we can observe an unbalanced ratio markedly in favor of *Firmicutes*, with a decrease in the percentage of *Bacteroidetes*. During the initial years of life, the microbiome of young children undergoes significant transformations, eventually evolving into a more mature configuration by the age of 3 or 4, at which point the rate of alteration decreases until after adulthood, where the microbiome seems to remain stable for extended periods, potentially throughout the entire lifespan [128]. Regarding the influence of diet on the gut microbiota, proteins from sources like red meat and dairy products have been shown to increase the population of biliary anaerobes and, notably, boost the abundance of *Bacteroides* [129]. Moreover, the fermentation of animal proteins can decrease the levels of *Bacteroides* and the production of SCFAs, triggering an inflammatory response in the intestines and potentially worsening chronic diseases. Nevertheless, consuming certain animal or plant proteins can have beneficial effects on the gut microbiota by promoting the growth of symbiotic microbiota, such as *Bacteroides* and *Lactobacillus*, in the intestines [130]. Under healthy circumstances, the gut microbiota demonstrates stability, resilience, and a symbiotic relationship with the host. Various intrinsic factors, like intestinal permeability, pH levels, and mucus production, may impact the composition and abundance of microbial communities. Presently, the predominant perspective suggests that a healthy gut microbiota community typically displays high taxonomic diversity, richness in microbial gene, and a consistent core microbiota composition (Figure 2).

With the rapid advancement of technologies like high-throughput sequencing, there has been an increased focus on understanding the functional aspects of the microbiome. The technology of next-generation sequencing (NGS) has enabled the understanding of the human microbiome and the characterization of unculturable microbes and their function. It is possible to sequence gut microbiota in the same time of different patients with some NGS methods, such as parallel 16S rRNA sequencing, shotgun metagenomic sequencing, and RNA sequencing. NGS facilitates the identification of more unique species than traditional culture methods and can determinate the relative abundance with the total amount of reads for each Phila to comprehend population heterogeneity. The most common NGS method in use is 16S rRNA with amplicon sequencing. This kind of sequencing involves first amplifying a region of the DNA via PCR, and then sequencing the resultant product. The target for PCR amplification is, most commonly, the bacterial 16S ribosomal RNA (rRNA) gene. The 16S rRNA gene serves as an ideal target due to its high conservation and widespread presence across bacteria; without it, bacteria would be incapable of translating mRNA into proteins, rendering them nonfunctional. Additionally, the gene encompasses nine hyper-variable regions (V1–V9) that vary among bacterial species and genera. However,

it is crucial to acknowledge that no single region can sufficiently differentiate all bacteria, and sequencing specific hypervariable regions may lead to varying interpretations of data. For instance, amplifying certain hypervariable regions could introduce biases in results, potentially causing under- or over-representation of taxa, although it could also aid in distinguishing between specific species within a genus. Recently, next-generation sequencing (NGS) of the entire 16S rRNA gene has emerged, and with increasingly sophisticated analytical techniques, it may offer both species and strain-level resolution in microbiota communities. Progress in genome sequencing has enabled researchers to explore the microbiota and their functions. The evidence gathered suggests that, although certain elements of the microbiota remain stable, the active members differ along the gastrointestinal tract, across different age demographics ranging from infants to the elderly, among indigenous communities to contemporary societies, and across various health states. Despite the ever-changing nature of the gut microbiota, it serves essential roles in the immunological, metabolic, structural, and neurological aspects of the human body. Consequently, the gut microbiota significantly impacts both the physical and mental well-being of individuals. Additionally, considering that metabolites produced by certain gut bacteria can serve as substrates or regulators of DNA methylation processes, it is conceivable that the epigenetic effects of early postnatal nutrition stem from alterations in the gut microbiota during this period. Interestingly, a recent study demonstrated that the gut microbiota can induce changes in the methylome of intestinal epithelial cells during postnatal development [131]. Further research is required to delve into the intriguing interplay between prenatal and postnatal nutrition, the infant gut microbiota, and epigenetic programming. The interaction between gut microbiota, dietary patterns, and epigenetic mechanism is of paramount importance for the host's health.

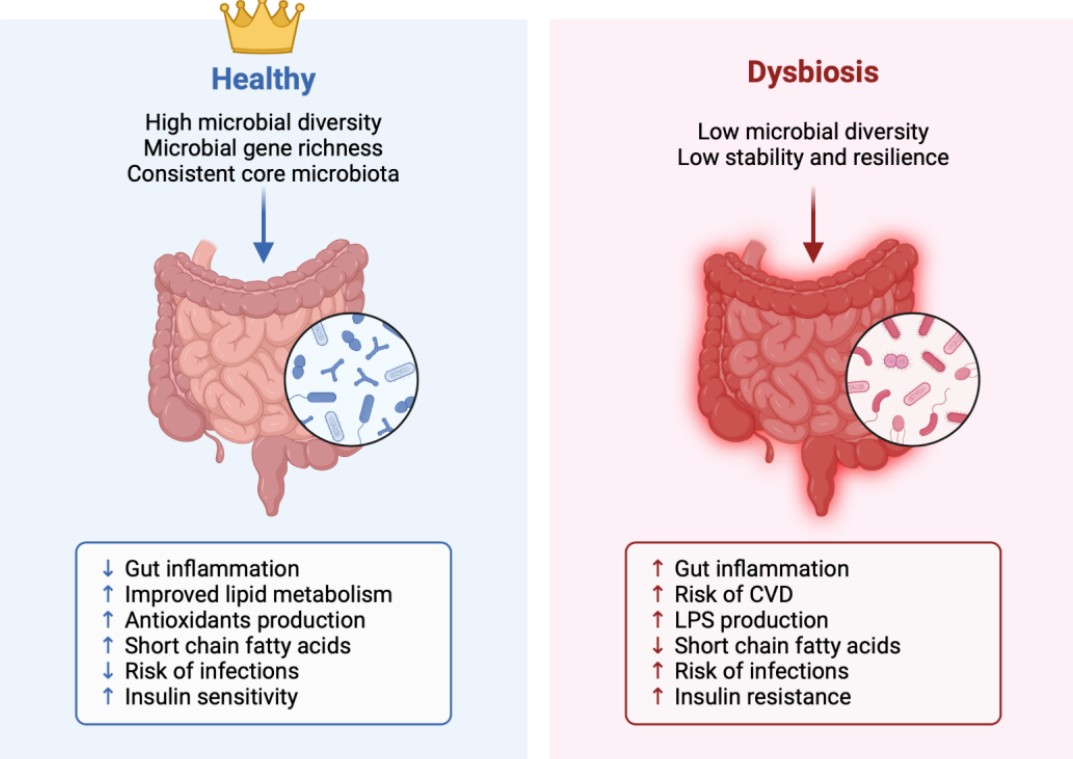

**Figure 2.** The role of gut microbiota in healthy patients and in patients with dysbiosis: healthy microbiota confers stability, resilience, and symbiotic interaction with the host due to high taxonomic diversity, microbial gene richness, and a consistent core microbiota. Differently, the dysbiosis of the gut microbiota is associated with the pathogenesis of both intestinal and extra-intestinal disorders.

## 6. Prevention Strategies and Epigenetics

A poor maternal diet is considered to be an adverse factor that affects the intrauterine environment [132] during critical periods of fetal developmental provoking lasting adverse effects on offspring health, such as type 2 diabetes and an increased risk for obesity [133]. Nonetheless, it has been reported that the offspring's metabolic profile and lipid metabolism can be boosted by inositol, resveratrol, and exercise in the mother, with an improvement in leptin, triglycerides, adiponectin levels, and a decrease in insulin resistance [134].

Moreover, bioactive food components may cause beneficial epigenetic changes at any time in life, with early nutrition being especially crucial. Folate, a water-soluble B vitamin, provides a one-carbon source for the synthesis of S-Adenosyl methionine, or AdoMet, which is required for DNA methylation, connecting the folate metabolism to phenotypic alterations through DNA methylation. Gene expression may be impacted also by other methyl donor nutrients, like choline, which can equally change the methylation status of DNA. Early in pregnancy, the availability of methyl donor nutrients on behalf of the mother is of vital importance for the correct development of the fetus, with long-term effects on the children's health and an increased susceptibility to disease, including cancer [135].

Although epigenetic targets for physical activity are mostly unknown, it has also been suggested that maternal exercise, in addition to a nutrition intervention, may alter fetal birth weight through epigenetic mechanisms [136]. Remarkably, maternal exercise totally offsets the negative effects of a high-fat-enriched diet on the offspring's metabolism [137] (Figure 3).

Following weaning, Lillycrop et al. examined the impact of an unbalanced maternal diet on the methylation status and the expression of the glucocorticoid receptor (GR) and peroxisomal proliferator-activated receptor (PPAR) genes in the livers of the rat offspring [138]. Throughout their pregnancies, the dams were fed a diet containing either a control protein (C), a restricted protein (R), or a restricted protein plus 5 mg/kg of folic acid (RF). It was reported that the PPAR gene methylation was found to be significantly lower and its expression levels significantly higher in the livers of the R compared to the C pup group. The methylation of the GR gene was also estimated to be significantly lower, while its expression was found to be higher in the R pups compared to the C ones. The RF diet, on the other hand, prevented these changes. No differences were detected in the methylation status and expression of the PPAR gene between groups. According to these findings, a prenatal unbalanced nutrition can cause long-lasting, gene-specific epigenetic alterations able to impact mRNA expression. However, folic acid supplementation may shield the offspring from these modifications in hepatic gene expression. This also suggests that therapeutic approaches to boost methyl group availability may be able to mitigate or even reverse the effects of early life environmental insults.

Maternal resveratrol supplementation has also been studied in the brain-isolated hippocampi of a directly exposed F1 generation and the transgenerational F2 generation in murine models [139]. The offspring originated from female senescence accelerated mouse-prone (SAMP8) mice that were given a diet rich in resveratrol for two months before they mated. It was observed that there had been a notable rise in global DNA methylation and a decrease in hydroxymethylation in F1 and F2. Subsequently, Tet2 and Dnmt3a/b gene expression levels were equally altered. Methylation levels in the promoters of the Nrf2 and NF-kβ genes were found to be increased in the offspring group, altering target gene expression and levels of hydrogen peroxide. The offspring of dams subjected to resveratrol supplementation exhibited elevated AMPKα activation, mTOR inhibition, and elevated levels of Beclin-1 protein and Pgc-1α gene expression. Thus, it could be assumed that, through epigenetic modifications and cell signaling pathways, maternal resveratrol supplementation could prevent cognitive impairment in the offspring of SAMP8 mice.

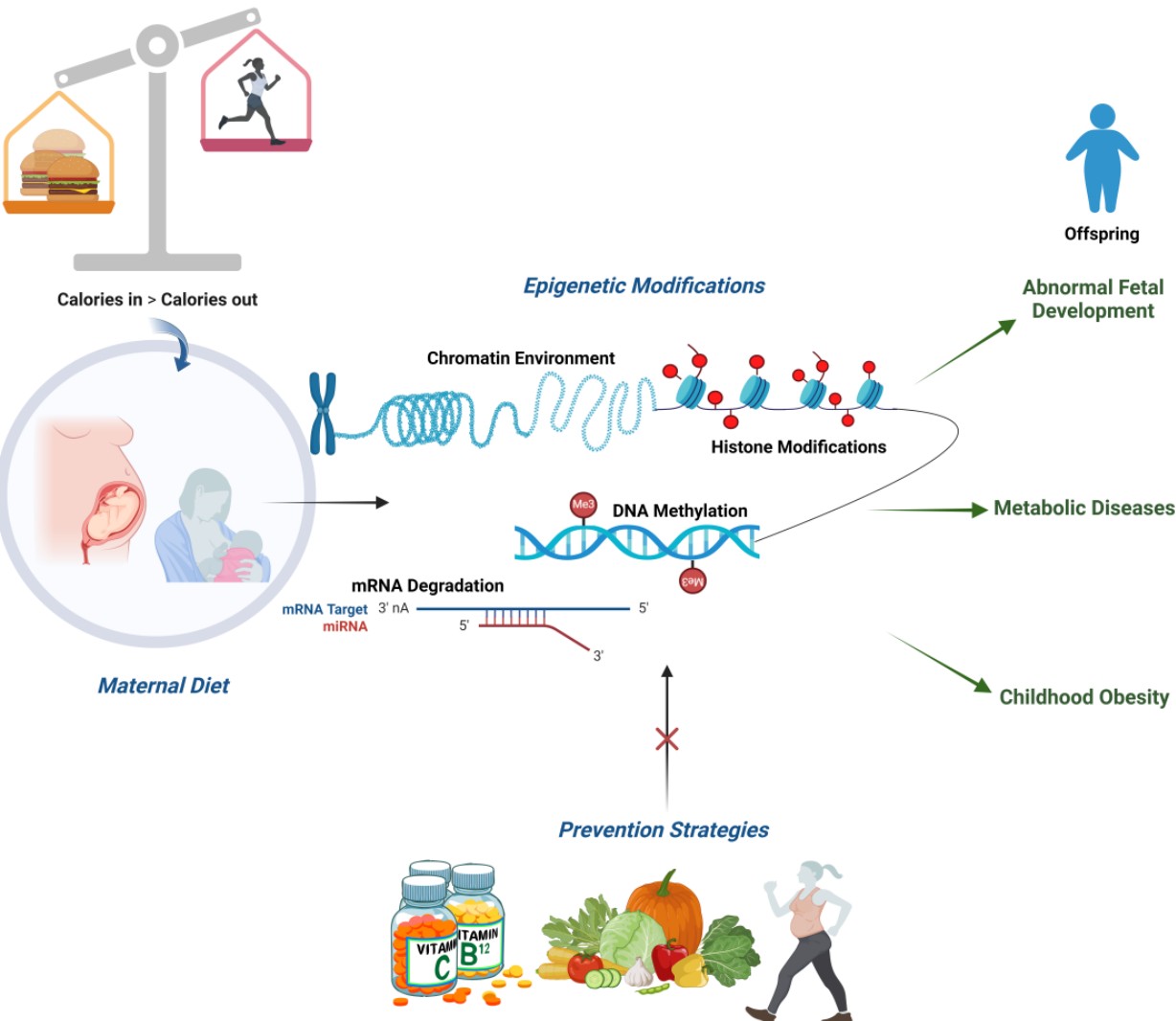

**Figure 3.** Schematic overview of the epigenetic modifications and health complications in the offspring due to an unbalanced prenatal and postnatal maternal diet, as well as the possible prevention strategies cancelling out this effect.

In addition, a previous study [140], which enrolled women and their offspring from clinic logs of five prenatal clinics in Durham County (USA), aimed to evaluate existing associations between prenatal physical activity (PA), birth weight, and DNA methylation levels in the newborn, targeting four imprinted genes, H19, MEG3, SGCE/PEG10, and PLAGL1, at differentially methylated regions (DMRs) that are known to be crucial for embryonic development. Peripheral blood samples were collected from the mothers and samples of the infants' cord blood at birth. Bisulfite pyrosequencing was used to determine the imprinted gene methylation levels at DMRs in 484 term mother-infant pairs. It was discovered that infants born to mothers in the highest quartile of total non-sedentary time had lower birth weights than those born to mothers in the lowest quartile. The strongest correlations were made among male newborns and, following adjustment for race/ethnicity, folic acid intake, gestational age at delivery, and maternal smoking, total non-sedentary time and birth weight was correlated with methylation at the PLAGL1 DMR ($p < 0.05$). Nonetheless, understanding in which ways prenatal PA may alter later health effects may require more investigation into the long-term consequences of alterations at the PLAGL1 DMR. Thus, the confirmation of similar results could offer information about the mechanisms behind the relationships between maternal PA and birth weight, as well as epigenetic targets for monitoring or treatment.

Preclinical studies have also been performed to highlight how maternal exercise could represent a potentially effective intervention to decrease the inheritance of multigenerational metabolic dysfunction provoked specifically by maternal obesity. Maternal exercise can have a variety of positive effects on the health of the offspring, one of which is the enhanced glucose metabolism due to DNA demethylation of important hepatic genes. In this context, a chow or high-fat diet was administered to virgin female C57Bl/6 and placenta-specific Sod3-knockout (Sod3f/f) mice for two weeks prior to conception, as well as during gestation, which were subsequently divided into two subgroups, trained and sedentary [141]. Offspring, on the other hand, were fed chow from birth onward. DNA methylation and biochemical levels were evaluated in the livers of the offspring of dams which were sedentary chow-fed, trained chow-fed, sedentary HFD-fed or trained HFD-fed. It was evidenced that a HFD caused a dysregulation of the offspring's liver glucose metabolism in C57BL/6 mice. This dysregulation was linked to an increased reactive oxygen species production, WD repeat-containing 82 (WDR82) carbonylation and inactivation of histone H3 lysine 4 (H3K4) methyltransferase, which resulted in a decrease in H3K4me3 at the promoters of genes involved in glucose metabolism. Interestingly, in the case of exercise during pregnancy on behalf of the HFD-fed dams, the entire signal was restored. Hepatoblasts overexpressing WDR82 replicated the effects of maternal exercise on H3K4me3 levels. Placental superoxide dismutase 3 (SOD3) was also found to be required for the regulation of H3K4me3, gene expression, and glucose metabolism, but N-acetylcysteine antioxidant therapy was not. In conclusion, it could be affirmed that maternal exercise may control a multicomponent epigenetic system in the developing liver of the fetus, passing on the health benefits of exercise to their offspring.

Nowadays, it is well-established that dietary and physical activity interventions during the gestational period can be linked to epigenome modifications in fetal tissues. However, further large-scale studies are of vital importance in order to ascertain the biological significance of these alterations.

## 7. Conclusions and Prevention Strategies for Future Health

Maternal metabolic and endocrine alterations modify the intrauterine environment, modulating nutrients for the fetus and predisposing the offspring to metabolic and non-metabolic diseases. This flow of transgenerational transmission is determined through two steps: the first involves the direct impairment of fetal tissues due to congenital alterations, such as cardiac hypertrophy, hyper/hypoinsulinemia, and the excessive deposition of hepatic lipids; the second includes the consequences of fetal programming, which occurs through cellular and molecular mechanisms, including the release of inflammatory cytokines, oxidative stress, protein modification and endocrine dysregulation. All of these alterations can trigger future events that predispose offspring to disease.

Longitudinal studies regarding memory-based mechanisms activated in utero that contribute to disease susceptibility are still lacking. The identification of such cellular mechanisms and critical time points of their activation is essential to develop new approaches to prevent fetal programming of disease in specific time windows.

One widely studied mechanism which could be potentially found behind disease susceptibility is represented by alterations in the epigenetic signature.

The perinatal period is a moment of rapid physiological changes, including epigenetic programming. Lifestyle choices can induce epigenetic modifications with significant consequences on systemic health and disease state. The nutrition of infants in the first 1000 days of life, from conception up to two years of age, has both immediate and long-term health consequences in terms of chronic non-communicable conditions.

The intrauterine environment is mainly regulated by maternal status, such as nutrition and placental function.

Several studies have evidenced that the adverse intrauterine environment can cause epigenetic changes throughout fetal development, which may persist into adult life.

It has been suggested that the strong correlation between nutrition and epigenetic changes could be explained by alterations in the one carbon cycle, which is fundamental for the availability of methyl groups, due to an unbalanced diet.

At least four distinct levels may be affected by the exposome-induced epigenetic modification: (1) fertility impairment due to alterations in the gametes' competence, (2) embryo development alterations, (3) the poor outcome of the assisted reproductive technology (ART) protocols and (4) a risk of developing pathologies in the adult life for the offspring [142].

Moreover, variations in dietary habits can lead to alterations in the of composition and the epigenetic status of microbial communities.

At present, dietary interventions, such as low dietary fat intake (<35%) with adequate fatty acid intake during gestation, represent the most effective intervention strategy to improve the maternal metabolic environment during pregnancy. Bioactive food components can also cause beneficial epigenetic changes at any time in life, with early nutrition being especially crucial. Therefore, it is essential to promote healthy and correct nutrition in women both in the pre-conceptional and post-conceptional phases in order to protect the newborn from the development of non-communicable diseases throughout life. Probiotic supplementation could mitigate insulin resistance and reduce the risk of obesity and diabetes. However, larger studies are needed to evaluate optimal dosage, frequency, and timing of supplementation, as well as safety and long-term effects on maternal, neonatal, and childhood outcomes.

**Author Contributions:** Conceptualization, M.F.F. and V.G.; literature search and first draft writing: F.U., L.A.M., F.A. and F.K., review and editing: M.F.F. and V.G.; supervision: L.S. All authors have read and agreed to the published version of the manuscript.

**Funding:** This research received no external funding.

**Conflicts of Interest:** The authors declare no conflicts of interest.

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
