# Peer review of "Exploring Maternal Diet-Epigenetic-Gut Microbiome Crosstalk as an Intervention Strategy to Counter Early Obesity Programming"

_cimb, doi:10.3390/cimb46050265_

Round 1

Reviewer 1 Report

Comments and Suggestions for Authors

This review examines some of the mechanisms behind fetal programming of adult diseases in the context of maternal under- or over-nutrition. After a summary of the maternal metabolic pathways that are altered in the context of maternal physiology, and how this interacts with the placenta and nutritional transport to the fetus, the review focuses mainly on examples of nutritional modulation of epigenetic changes altering the expression of key genes in the mother and fetus. It then extends the concept that nutritionally-driven changes in the maternal microbiome can modulate epigenetic patterning of genes to impact fetal growth trajectories, and how this represents risk of adult metabolic diseases in the offspring. This linkage of emerging evidence and concepts is novel and represents a thought-provoking review of an emerging area of science. I enjoyed reading it.

Perhaps one of the biggest limitations of the review is that in both the areas of nutritional regulation of epigenetic changes in metabolism-associated hormones, and in the role of the microbiome, there is still not a large evidence base in human. The review, I think by design, focuses mostly on human biology and only occasionally refers to animal models. A larger extrapolation of animal models may generate a wider range of known target genes and involved organs.

The microbiome section is very reliant on associations between microbiome genotyping and clinical phenotype and child outcome. There is always the question of chicken vs. egg in such studies. Did the metabolic disorder change the microbiome, or vice versa? Hence, the key question is do interventions to modify the maternal microbiome actually prevent or reverse adverse epigenetic changes and change the phenotype of the fetus and child? Are there any studies from, say, fecal transplants that have shown that modification of the maternal microbiome has beneficial effects on maternal metabolism, fetal birth weight or childhood adiposity in situations of metabolic stress such as maternal obesity? Are there animal studies that would point to this being a therapeutic avenue of study?

Whilst excellent examples are given for epigenetic changes to the promoter regions of key genes known to influence fetal development, such as GCs, is there a literature that suggests changes in stem cell lineages in the embryo, or lineage commitment of progenitor cells in specific organs in fetal development? An example could be pancreas where restriction of nutrient availability to fetal rodents changes the epigenetic control of PDX1 resulting in a deficiency of beta-cell mass at birth and impaired glucose-sensitive insulin release. In that model the epigenetic modifications to the PDX1 promoter progressively limit gene expression with advancing age in the offspring. When couple to epigenetic changes in adipocyte progenitors and altered insulin sensitivity in the results adipocytes, this results in the development of glucose intolerance in the adult offspring.

A more minor point, but in each section the descriptions of the implications of undernutrition vs. overnutrition in the mother seem to be covered in no particular order. It might help the reader if each section starts with the implications of undernutrition, followed by overnutrition (or the reverse) for consistency.

Author Response

This review examines some of the mechanisms behind fetal programming of adult diseases in the context of maternal under- or over-nutrition. After a summary of the maternal metabolic pathways that are altered in the context of maternal physiology, and how this interacts with the placenta and nutritional transport to the fetus, the review focuses mainly on examples of nutritional modulation of epigenetic changes altering the expression of key genes in the mother and fetus. It then extends the concept that nutritionally-driven changes in the maternal microbiome can modulate epigenetic patterning of genes to impact fetal growth trajectories, and how this represents risk of adult metabolic diseases in the offspring. This linkage of emerging evidence and concepts is novel and represents a thought-provoking review of an emerging area of science. I enjoyed reading it.

Q1. Perhaps one of the biggest limitations of the review is that in both the areas of nutritional regulation of epigenetic changes in metabolism-associated hormones, and in the role of the microbiome, there is still not a large evidence base in human. The review, I think by design, focuses mostly on human biology and only occasionally refers to animal models. A larger extrapolation of animal models may generate a wider range of known target genes and involved organs.

A1. We thank the reviewer for the observation. Future human studies regarding a variety of maternal dietary habits before, throughout and even after pregnancy in order to evaluate the microbiome and potential modifications in the epigenetic signature of the offspring are of vital importance and should be performed on large scale especially considering the already existing literature on the subject, which at the moment is not indeed largely human-based. For this reason, one of our goals was to collect all possible clinical information and better comprehend what is currently known and which should be the next steps in long-term prospective, preclinical studies and new clinical trials, for example, in humans. Undoubtedly, the topic would not be complete without mentioning some very well-conceived studies focused on animal models, especially regarding prevention strategies able to cause beneficial epigenetic changes at any time in life, which were evaluated and included in the submitted version of our manuscript.

Q2. The microbiome section is very reliant on associations between microbiome genotyping and clinical phenotype and child outcome. There is always the question of chicken vs. egg in such studies. Did the metabolic disorder change the microbiome, or vice versa? Hence, the key question is do interventions to modify the maternal microbiome actually prevent or reverse adverse epigenetic changes and change the phenotype of the fetus and child? Are there any studies from, say, fecal transplants that have shown that modification of the maternal microbiome has beneficial effects on maternal metabolism, fetal birth weight or childhood adiposity in situations of metabolic stress such as maternal obesity? Are there animal studies that would point to this being a therapeutic avenue of study?

A2. We thank the reviewer for the opportunity to better clarify this aspect. As pointed out, a series of studies have been carried out about possible modifications of the maternal microbiome in, for example, fecal transplants, which suggest that these kinds of interventions may reverse adverse epigenetic changes. In order to include this point of view, we have additionally slightly modified the paragraph 5, citing, initially, an important experimental study explaining how the FMT (Fecal Microbiota Transplant) could have a beneficial effects on fetal gut microbiota. A related reference has been added as follows:

“Given that it seems probable that the crucial bacteria obtained by infants born vaginally originate from the maternal fecal pool, one method to establish a healthy bacterial population in the infant gut following cesarean delivery has been through Fecal Microbiota Transplantation (FMT) using the mother as a donor. In a study of Korpela et al. they evaluate the FMT approach in 17 mothers. This approach has been demonstrated to furnish infants delivered via cesarean section with microbiomes similar to those observed in infants born vaginally, highlighting the importance that the most pivotal period for the development of the human microbiome occurs during the initial 2–3 years of life. Addressing dysbiosis during pregnancy and early childhood can profoundly impact the future health of an infant.”

Also, some animal studies that have provided approaches to modify the microbiome result being of particular interest. In particular, a study performed in a canine model stood out, where the FMT managed to improve the fecal bacterial diversity, with a metabolome more closely resembling that of a healthy control. A related reference has been equallt added as follows:While the therapeutic efficacy of Fecal Microbiota Transplantation (FMT) has been well established in humans through various studies, its application in animal models, particularly in dogs, has been less explored. Only a few studies have investigated the use of FMT in dogs. One of these studies highlighted the benefits of FMT, demonstrating increased fecal bacterial diversity and a metabolome more closely resembling that of healthy controls. Additionally, FMT shows potential as a therapeutic approach for dogs suffering from chronic intestinal diseases and related dysbiosis, such as various chronic enteropathies and exocrine pancreatic insufficiency. These evidences could suggest that interventions to modify the maternal microbiome could actually prevent adverse epigenetic changes and modify the phenotype of the fetus. By these studies, fecal transplants have shown that improvements in the maternal microbiome have beneficial effects on maternal metabolism and on fetal birth weight and microbial diversity”.

Q3. Whilst excellent examples are given for epigenetic changes to the promoter regions of key genes known to influence fetal development, such as GCs, is there a literature that suggests changes in stem cell lineages in the embryo, or lineage commitment of progenitor cells in specific organs in fetal development? An example could be pancreas where restriction of nutrient availability to fetal rodents changes the epigenetic control of PDX1 resulting in a deficiency of beta-cell mass at birth and impaired glucose-sensitive insulin release. In that model the epigenetic modifications to the PDX1 promoter progressively limit gene expression with advancing age in the offspring. When couple to epigenetic changes in adipocyte progenitors and altered insulin sensitivity in the results adipocytes, this results in the development of glucose intolerance in the adult offspring.

A3. We thank the reviewer for the kind suggestion. Our review’s main focus was to consider the crosstalk relationship between the maternal diet, gut microbiome and possible underlying epigenetic modifications, both at a pre- and postnatal level, potentially affecting fetal development and, subsequently, obesity programming in the offspring. Thus, our searches involved reproductive tissues and cells, such as the placenta, as well as whole blood and buccal epithelial cells based on existing literature. Nonetheless, changes can undoubtedly occur also in organs such as pancreas, where, for example, a restriction of nutrients availability due to a low-protein maternal diet during gestation can affect the expression of key pancreatic β-cell genes, like Pdx1, and the methylation status of regulatory regions of genes, such as MafA, in the offspring of Wistar rats (Sosa-Larios TC et al., 2023). This process, although not associated with obesity programming, may contribute to developmental dysregulation of β-cell function and influence the long-term health of the offspring. For this reason, a related reference has been added in paragraph 4 (Maternal Dietary Factors Influencing Epigenetic Changes) as follows:

“As it has been shown in the pancreas of murine models, a low-protein maternal diet during pregnancy may limit the availability of nutrients, leading to alterations in the expression and methylation status of key pancreatic genes, such as Pdx1 and MafA, and to consequent developmental dysregulation of β-cell activity with long-term health consequences on the offspring”.

Q4. A more minor point, but in each section the descriptions of the implications of undernutrition vs. overnutrition in the mother seem to be covered in no particular order. It might help the reader if each section starts with the implications of undernutrition, followed by overnutrition (or the reverse) for consistency.

A4. We thank the reviewer for pointing this out. Throughout our manuscript’s main text a neat subdivision between the concepts of undernutrition and overnutrition has not been entirely made for a variety of reasons.  For example, in the section focused on the epigenetic changes due to maternal dietary factors, the paragraph was structured based on the prenatal and postnatal exposure of the offspring to compromised maternal nutritional patterns and therefore the time period of the exposure resulted the determining factor of such division. Other conceptual divisions also have been, where papers citing genes involved in metabolic processes, such as leptin, were involved and therefore inserted consecutively. As far as the prevention strategies are concerned, a division between literature focused on antioxidant supplementation and that centred around maternal exercise was also preferred. Nonetheless, there are parts where a distinction between overnutrition, due to, for instance, high fat and high-sugar diets or obesity, and undernutrition in utero is predominant, such as paragraph 2, dedicated on the maternal nutrition and its effects on fetuses and newborns, offering a baseline for primal introducing information on the topic.

Reviewer 2 Report

Comments and Suggestions for Authors

This review is an excellent  update on the role of epigenetics in programming child and maternal health during gestation and lactation.

Missing is the influence of maternal or grand-maternal status on programming and the transgenerational aspect. (Vickers MH. Developmental programming and transgenerational transmission of obesity. Ann Nutr Metab. 2014;64 Suppl 1:26-34. doi: 10.1159/000360506. Epub 2014 Jul 23. PMID: 25059803.)

Nutrition and pre-gestational BMI must be addressed. (Han SM, Derraik JGB, Vickers MH, Devaraj S, Huang F, Pang WW, Godfrey KM, Chan SY, Thakkar SK, Cutfield WS; NiPPeR Study Group. A nutritional supplement taken during preconception and pregnancy influences human milk macronutrients in women with overweight/obesity and gestational diabetes mellitus. Front Nutr. 2023 Oct 17; 10:1282376. doi: 10.3389/fnut.2023.1282376. PMID: 37915619; PMCID: PMC10616264.)

Nutrition for pregnant and breastfeeding women is sorely lacking in the direct role it plays in the programming of the child and the mother, and in practical nutritional advice and/or supplements.(

1-       Carretero-Krug A, Montero-Bravo A, Morais-Moreno C, Puga AM, Samaniego-Vaesken ML, Partearroyo T, Varela-Moreiras G. Nutritional Status of Breastfeeding Mothers and Impact of Diet and Dietary Supplementation: A Narrative Review. Nutrients. 2024 Jan 19;16(2):301. doi: 10.3390/nu16020301. PMID: 38276540; PMCID: PMC10818638.

2-       Norrish I, Sindi A, Sakalidis VS, Lai CT, McEachran JL, Tint MT, Perrella SL, Nicol MP, Gridneva Z, Geddes DT. Relationships between the Intakes of Human Milk Components and Body Composition of Breastfed Infants: A Systematic Review. Nutrients. 2023 May 18;15(10):2370. doi: 10.3390/nu15102370. PMID: 37242254; PMCID: PMC10223764.

3-       Ureta-Velasco N, Montealegre-Pomar A, Keller K, Escuder-Vieco D, Fontecha J, Calvo MV, Megino-Tello J, Serrano JCE, García-Lara NR, Pallás-Alonso CR. Associations of Dietary Intake and Nutrient Status with Micronutrient and Lipid Composition in Breast Milk of Donor Women. Nutrients. 2023 Aug 7;15(15):3486. doi: 10.3390/nu15153486. PMID: 37571421; PMCID: PMC10421487.)

The role of bioactive elements in breast milk, such as HMOs and the mother's Lewis or non-Lewis secretory status, is not addressed, even though this is an important factor in the nature of the maternal and neonatal microbiota.(  

1-Abisi HK, Kabahweza HM, Okutoyi L, Wamalwa DC, Nduati RW. The Role of Maternal Secretor Status and Human Milk Oligosaccharides on Early Childhood Development: A Systematic Review and Meta-Analysis. Breastfeed Med. 2024 Apr 5. doi: 10.1089/bfm.2023.0274. Epub ahead of print. PMID: 38577928.

2- Arzamasov AA, Osterman AL. Milk glycan metabolism by intestinal bifidobacteria: insights from comparative genomics. Crit Rev Biochem Mol Biol. 2022 Oct-Dec;57(5-6):562-584. doi: 10.1080/10409238.2023.2182272. Epub 2023 Mar 3. PMID: 36866565; PMCID: PMC10192226.

4-       Ouyang R, Ding J, Huang Y, Zheng F, Zheng S, Ye Y, Li Q, Wang X, Ma X, Zou Y, Chen R, Zhuo Z, Li Z, Xin Q, Zhou L, Lu X, Ren Z, Liu X, Kovatcheva-Datchary P, Xu G. Maturation of the gut metabolome during the first year of life in humans. Gut Microbes. 2023 Jan-Dec;15(1):2231596. doi: 10.1080/19490976.2023.2231596. PMID: 37424334; PMCID: PMC10334852.)

I suggest that the authors consider that the role of epigenetics, as they have pointed out, is reversible, provided that they consider the direct effect of maternal nutrition, pregestational-gestational and especially postnatal, with the role of the nutritional and bioactive components of breast milk (HMOs, MFGM, LCPUFA, Microbiome with their own regulation by nutrients and especially by milglycans).

But also, as emphasized by lifestyle.

Author Response

Reviewer 2

This review is an excellent update on the role of epigenetics in programming child and maternal health during gestation and lactation.

Q1. Missing is the influence of maternal or grand-maternal status on programming and the transgenerational aspect. (Vickers MH. Developmental programming and transgenerational transmission of obesity. Ann Nutr Metab. 2014;64 Suppl 1:26-34. doi: 10.1159/000360506. Epub 2014 Jul 23. PMID: 25059803.)

A2. we reported in the paragraph 3 the influence of maternal status on fetal programming and the suggested reference (n. 43)

 Q2. Nutrition and pre-gestational BMI must be addressed. (Han SM, Derraik JGB, Vickers MH, Devaraj S, Huang F, Pang WW, Godfrey KM, Chan SY, Thakkar SK, Cutfield WS; NiPPeR Study Group. A nutritional supplement taken during preconception and pregnancy influences human milk macronutrients in women with overweight/obesity and gestational diabetes mellitus. Front Nutr. 2023 Oct 17; 10:1282376. doi: 10.3389/fnut.2023.1282376. PMID: 37915619; PMCID: PMC10616264.)

A2: we extensively reported both in introduction and in paragraph 2 the effects of maternal nutrition and BMI on the fetus and newborn.

Q3. Nutrition for pregnant and breastfeeding women is sorely lacking in the direct role it plays in the programming of the child and the mother, and in practical nutritional advice and/or supplements.

A3. The topic of nutrition and breastfeeding is very interesting and a lot of recent reviews have been published as reported by the reviewer. This was not the focus our review. However, we reported in the paragraph 4 this issue and the reference that summarize the aspects highlighted by the reviewer (see reference number 86)

Q4. The role of bioactive elements in breast milk, such as HMOs and the mother's Lewis or non-Lewis secretory status, is not addressed, even though this is an important factor in the nature of the maternal and neonatal microbiota.  

A4. we reported in the paragraph 4 the sentence referred to this issue and the reference number 86.

Q5. I suggest that the authors consider that the role of epigenetics, as they have pointed out, is reversible, provided that they consider the direct effect of maternal nutrition, pregestational-gestational and especially postnatal, with the role of the nutritional and bioactive components of breast milk (HMOs, MFGM, LCPUFA, Microbiome with their own regulation by nutrients and especially by milglycans). But also, as emphasized by lifestyle.

A5. Again, this was not the focus of our review, we reported in the paragraph 4 a sentence referred to this issue and the reference number 86.
